# Utilization of Viral Vector Vaccines in Preparing for Future Pandemics

**DOI:** 10.3390/vaccines10030436

**Published:** 2022-03-12

**Authors:** Kimberly A. Hofmeyer, Katherine M. Bianchi, Daniel N. Wolfe

**Affiliations:** US Department of Health and Human Services, Office of the Assistant Secretary for Preparedness and Response, Biomedical Advanced Research and Development Authority, Washington, DC 20201, USA; kimberly.hofmeyer@hhs.gov (K.A.H.); katherine.bianchi@hhs.gov (K.M.B.)

**Keywords:** vaccines, vectors, virus, pandemic, clinical trials

## Abstract

As the global response to COVID-19 continues, government stakeholders and private partners must keep an eye on the future for the next emerging viral threat with pandemic potential. Many of the virus families considered to be among these threats currently cause sporadic outbreaks of unpredictable size and timing. This represents a major challenge in terms of both obtaining sufficient funding to develop vaccines, and the ability to evaluate clinical efficacy in the field. However, this also presents an opportunity in which vaccines, along with robust diagnostics and contact tracing, can be utilized to respond to outbreaks as they occur, and limit the potential for further spread of the disease in question. While mRNA-based vaccines have proven, during the COVID-19 response, to be an effective and safe solution in terms of providing a rapid response to vaccine development, virus vector-based vaccines represent a class of vaccines that can offer key advantages in certain performance characteristics with regard to viruses of pandemic potential. Here, we will discuss some of the key pros and cons of viral vector vaccines in the context of preparing for future pandemics.

## 1. Introduction

As of the end of January 2022, there have been over 373 million diagnosed cases of COVID-19, with more than 5.65 million deaths worldwide [1]. These numbers have unfortunately underscored key gaps in preparedness for pandemic threats. As the global response to COVID-19 continues, public and private partners must also keep an eye on the future to improve our collective preparedness posture for future pandemics.

There are several viral threats identified by the World Health Organization (WHO) as representing major public health risks due to their potential to cause epidemics and pandemics. These include severe acute respiratory syndrome (SARS) coronavirus (CoV) 2, or SARS-CoV-2, Middle East respiratory syndrome coronavirus (MERS-CoV), SARS-CoV, Crimean-Congo hemorrhagic fever virus, Ebola viruses, Marburg virus, Lassa virus, Nipah virus, Hendra virus, Rift Valley fever virus, Zika virus, and “disease X” [2]. Similar lists of priority viral threats have been published by GAVI and CEPI on their respective websites, which include these threats, plus Chikungunya virus and monkeypox virus [3,4]. While COVID-19 has had an obvious global impact both in terms of lives lost and economic disruption, there is a collective threat of viral pathogens with pandemic potential, representing viruses that could be particularly problematic due to their ability to rapidly spread and cause severe disease, morbidity, and mortality (Table 1). In some cases, the endemic nature of these diseases, whether in human or animal populations, adds to the risk of epidemic or pandemic disease.

Diagnostics and disease surveillance will play a critical role in stopping outbreaks early; however, there are key roles that vaccines can play in reducing disease burden in the context of outbreaks. While there is an understandable push to leverage “platform” technologies that could represent rapid plug-and-play solutions as part of a vaccine response, the most critical aspect is that the right vaccine technology is utilized for the threat in question. In some cases, such as Ebola and Lassa viruses, this may mean viral vector vaccines due to the ability to provide rapid onset to protection with a single dose. Historically, Ebola outbreaks have been sporadic and unpredictable in terms of size and geographic location [24]. As a result, the now-licensed Ebola vaccine has been utilized in response to outbreaks by vaccinating those at risk of exposure, since efficacy was demonstrated in 2015 [25,26]. The clinical operations associated with Ebola vaccine responses were made possible by a vaccine with high efficacy and a rapid onset of protection, two key performance parameters associated with the use of a live virus-vector vaccine in this case [26]. For Lassa virus, human and animal studies indicate that convalescence is largely mediated by the cellular response. Antibody titers are highest only after infected individuals have already recovered from disease [27]. Since cellular response plays a dominant role during resolution of natural infection, vaccine development for Lassa virus has included a focus on generating a potent CD8+ T cell response [27]. Because viral vectors are live, attenuated viruses, they generally elicit a robust cellular response, and will be an important platform option for viruses where this is a key to optimal protective immunity.

With the goal of pandemic preparedness in mind, it will remain critical that we find the right vaccine for each respective threat. For diseases of which outbreaks will continue to be sporadic, and geographically isolated at the start, there will be time to intervene with vaccines prior to the event becoming an epidemic or pandemic. Some of the advantages offered by vaccines utilizing virus vectors may provide opportunities to respond in a reactive setting if targeted vaccination campaigns can be used to slow the spread of disease. A recent review by Vrba et al. highlighted the breadth of viruses that have been used to express heterologous antigens as a means for delivery as a vaccine [28]. In this review, we will focus on those virus vectors that have proceeded to clinical development, and how these vectors could be applied to emerging viral threats of pandemic potential.

## 2. Target Product Profile

Preparedness for the threats noted above would be best served by developing a pipeline of candidate vaccines with key performance parameters in mind at the outset. Viral vector vaccines have positive aspects for several of the parameters noted in Table 2.

### 2.1. Safety and Efficacy

Several viral vector-based vaccines are approved or authorized for use, based on a favorable balance between the benefits of vaccine effectiveness, the seriousness of the threat, and a comparably favorable safety profile. The approved vesicular stomatitis virus-vector Ebola vaccine showed early indications of arthralgia and arthritis events during clinical development [26,29]. While these early findings altered safety monitoring plans, ultimately, these events occurred at a low frequency, and were resolved relatively quickly [29]. Being a live virus vaccine, the thorough characterization of biodistribution and viral shedding was also monitored [30].

Recent findings of rare thrombolytic events with two different adenovirus-vector COVID-19 vaccines underscore the importance of tailoring safety monitoring plans during clinical development, and continued safety monitoring post-authorization [31]. The rare yet serious adverse events associated with adenovirus vectors have resulted in several regulatory authorities amending their respective guidance on usage [32]. However, given the significant threat of COVID-19, the benefits of these vaccines, and the rarity of thrombolytic events, the vaccines remain authorized for use. If the benefit-risk profile warrants continued development, safety monitoring can be fine-tuned to address key questions associated with the vector in question.

A high level of efficacy can be critical with diseases with high morbidity and mortality rates, which may require a relatively reactogenic vaccine. Smallpox is a good example of a disease for which vaccines with higher levels of reactogenicity were acceptable given the high case fatality rates (~30%) and significant morbidity in survivors (e.g., blindness and skin scarring) [33]. Analyses from the United States vaccination efforts throughout the 1960s showed an adverse event profile that included progressive vaccinia, eczema vaccinatum, postvaccinial encephalitis, and generalized vaccinia at rates from 1.5 to 241 cases per million vaccinations [34]. As the global burden of smallpox disease declined over the 20th century, safer vaccines that use vaccinia vectors, but include extensive cell culture passaging to attenuate the vaccine, have been pursued [35].

### 2.2. Clinical Operations

Efficacy following a single dose can be a huge benefit when a rapid response is required, or when tracking individuals for subsequent doses presents a challenge. Specifically, when clinical operations may entail use in a ring vaccination setting, rapid onset to protection with a single dose is crucial to the ability to slow an outbreak. When combined with disease surveillance, a vaccine that elicits rapid protection against disease can be utilized to protect those likely to have been exposed or be exposed, such as contacts of index cases and healthcare workers. Protection with a single dose also confers advantages around the total number of doses needed, which can be a critical consideration in resource-constrained settings, not only because of the number of doses, but also the infrastructure to track needs for additional doses.

Cold-chain requirements are also important to consider, although responses to Ebola and SARS-CoV-2 have demonstrated that the distribution and administration of a vaccine requiring storage at −70 °C are feasible [36,37]. That said, a vaccine that can be stored long-term at refrigerated or room temperatures offers clear logistical advantages in storage, both in terms of central inventory, and at the point of use. Viral vector vaccines can be developed with such storage and shelf-life considerations in mind. Viral vectors are amenable to techniques such as lyophilization to improve storage and shelf-life considerations [38], although this can also introduce complications in the field with diluent requirements.

### 2.3. Research and Development

While not inherently a key vaccine characteristic, development speed will be a critical aspect in how quickly we can meet key performance parameters. Generally speaking, when considering the three major vaccine classes (nucleic acid-based, viral vector, and recombinant protein constructs) likely to be pursued, nucleic acid-based vaccines likely will be the quickest into clinical development, followed by virus vector vaccines, and then protein subunit-based vaccines.

Part of the speed associated with vaccine development will lie in whether the production platform has been applied to other pathogens. For example, the rapid progress of COVID-19 vaccines utilizing mRNA and adenovirus vector platforms was enabled by past research and development efforts using these same platforms for other pathogens [39]. For viral vector vaccines, experience in applying reverse genetics technologies to generate constructs and the conduct of nonclinical studies to understand toxicity and immunogenicity profiles associated with the vector provides a foundation on which candidate development can quickly progress. Examples of viral vectors with such experience will be discussed in the next section.

## 3. Virus Vectors in Development

### 3.1. Adenovirus Vectors

The utility of adenoviruses as vaccine vectors became clear when the earliest attempts at using them for gene therapy were hindered by the robust induction of innate and adaptive immune responses [40]. These findings, along with their broad cellular tropism, high gene expression, and manufacturing ease, have made adenoviruses one of the most widely used vaccine platforms. Adenoviruses are typically attenuated for use as vaccine vectors by deleting the E1/E3 genomic locus, which is replaced with a transgene encoding the target pathogen antigen. E1 deletion renders the virus non-replicating and E3 deletion removes proteins that would otherwise help the virus evade immune detection in infected cells. E1/E3 deleted adenovirus vectors can accommodate transgene inserts of up to 7.5–8 kb in size [40]. Table 3 highlights the use of various vectors against a range of infectious diseases.

Human adenovirus 5 (Ad5) was originally the most widely used adenovirus vector for vaccine development. Ad5 vector COVID-19 vaccines are approved as single shot vaccines in China (Ad5-nCoV), with an estimated 57.5% vaccine efficacy against symptomatic COVID-19 [51], and as part of a heterologous boost to an Ad26 prime in Russia (Sputnik V) [80].

The early clinical development of Ad5-based vaccines found that high seroprevalence for Ad5 within the human population reduced antigen-specific immunity for Ad5 vector HIV vaccines [81]. In response, the development of adenovirus vector vaccines has shifted to focus primarily on either human adenoviruses with low seroprevalence (e.g., human serotypes 26 (Ad26) and 35 (Ad35)) or primate adenoviruses [82,83].

Ad26 emerged as an attractive alternate to address the pre-existing immunity issues with Ad5, as it has low human seroprevalence, and is more immunogenic compared to other alternate adenovirus vectors [84]. Prior to the COVID-19 pandemic, Ad26 vector vaccines had been evaluated in several clinical trials for various pathogens, and were determined to have an acceptable safety profile, and induce strong and durable humoral and cellular immune responses [85]. The first approved Ad26 vector vaccine was authorized under exceptional circumstances by the European Medicines Agency (EMA) in 2020 for Ebola virus when combined with a heterologous boost of a modified vaccinia Ankara (MVA) vector-based vaccine known as Zabdeno and Mvabea [43]. This heterologous prime/boost regimen was well-tolerated and generated robust antibody responses in clinical trials [85].

An Ad26 vector vaccine for SARS-CoV-2 (Ad26.COV2.S) is one of three vaccines authorized by the US Food and Drug Administration (FDA) for COVID-19. In a Phase 3 trial, a single shot of the Janssen Ad26.COV2.S vaccine demonstrated an efficacy of 66.9% against moderate to severe–critical COVID-19, with an onset 14 days after vaccine administration [86]. In an analysis of Ad26.COV2.S immunogenicity against COVID-19 variants of concern, while some reduction in neutralizing antibodies was seen for variants versus the prototype SARS-CoV-2 strain, the CD8+ and CD4+ T cell responses remained comparable against all strains [87]. The induction of a CD8+ T cell response may prove to be important for controlling infections with COVID-19 variants if they can escape neutralization by vaccine- or infection-generated antibodies targeting the original strain.

Ad35 is another human adenovirus alternate to Ad5 that has lower seroprevalence in humans and is immunogenic, although less so than Ad26 [84,88]. While there are no clinical candidates for the viruses of interest discussed here that use Ad35, the AdCLD-CoV19 candidate uses an Ad5 vector in which the native fiber knob protein used for virus-cell attachment is replaced by the Ad35 counterpart [72]. This design hopes to take advantage of the high immunogenic potential of Ad5, while avoiding pre-existing immunity issues.

ChAdOx1 is a serotype Y25 chimpanzee adenovirus vector with deletion of the E1/E3 loci and additional modifications that replace regions of the native E4 loci with that of Ad5 to increase virus yields [89]. The use of a chimpanzee adenovirus vector circumvents the issues with pre-existing immunity to human Ad5, which is confirmed by studies that show a very low seroprevalence of neutralizing antibodies for Y25 in humans [89]. ChAdOx1 is in use for several clinical stage vaccine candidates for the viruses of interest (Table 3).

Following demonstration that a single dose of a ChAdOx1 MERS-CoV vaccine generated protective immunity in non-human primates (NHP), the candidate completed Phase 1 human trials, where it proved to be well tolerated, and elicited humoral and cellular responses [61,90]. These data informed the selection of this platform for the development of AZD1222 to target SARS-CoV-2. AZD1222 is well tolerated as a two-dose primary series, and in Phase 3 trials, demonstrated an overall estimated efficacy of 74.0% at 15 or more days after the second dose [91]. This vaccine received conditional marketing authorization from the EMA in early 2021 [60]. Similar to Ad26.COV2.S, in addition to generating neutralizing antibodies, AZD1222 also elicited spike antigen-specific CD4+ and CD8+ T cells [92].

Chimpanzee adenovirus type 3 (cAd3) was identified as one of the most immunogenic primate adenovirus vectors, based on early screens in animal models [83]. In clinical trials, a cAd3-based Ebola virus vaccine was well tolerated and able to induce humoral and cellular responses, either alone [93], or as a priming immunization for a MVA-BN-Filo boost [94]. cAd3 is also used in the clinical development of monovalent vaccines for two other filoviruses, Sudan virus and Marburg virus, as well as a bivalent vaccine for Ebola virus and Sudan virus [66,67,68,69].

Additional primate adenovirus vectors in clinical development include the chimpanzee adenoviruses serotype 68 (ChAd68) and serotype 36 (Ad36), as well as the gorilla adenovirus serotype 32 (GRAd32), which are all in Phase 1 trials for SARS-CoV-2 (Table 3). The Ad36 vector SARS-CoV-2 vaccine is being assessed in a trial as an intranasal vaccine, which previously generated both systemic and sterilizing mucosal immunity in small animal models [71,95].

Single cycle adenovirus vectors (SC Ad) do not produce viral progeny, but are different from all the above-mentioned non-replicating vectors, in that they can go through a single round of gene amplification. This is achieved by retaining the genes required to amplify the inserted antigen cassette, but deleting the late viral genes that are involved in viral assembly [96]. The result is a vector with a safety profile similar to other non-replicating adenovirus vectors, but with increased amplification of the antigen target gene, higher production of the antigen target protein, and increased immunogenicity [96]. There is currently a human SC Ad serotype 6 vector (SC Ad6) in clinical development for SARS-CoV-2 [70]. Ad6 has lower seroprevalence in humans than Ad5 and has proven to be immunogenic in preclinical animal studies [97].

### 3.2. Modified Vaccinia Ankara (MVA)

MVA was originally attenuated through the serial tissue culture passage of the vaccinia virus strain Ankara. This resulted in genome deletions that restricted the host range of MVA, thus rendering the vector non-replicating in human cells, and removed immunomodulatory genes [98]. In turn, MVA has a long history as a safe and effective vaccine, as is evidenced by the recent approval of MVA for smallpox and monkeypox, and its prior use in the later stages of the World Health Organization’s (WHO) smallpox eradication campaign [35].

As an orthopoxvirus, MVA has a particularly large genome that can accommodate the insertion of transgenes of up to 25–30 kB in size [28]. This has made MVA an attractive vector to use in multivalent vaccines that target several pathogens at a time. The multivalent MVA Filovirus vaccine (MVA-BN Filo or Mvabea) has a transgene insert encoding for the surface glycoproteins (GP) of Ebola virus, Sudan virus, and Marburg virus, as well as for the nucleoprotein of Tai Forest virus [99]. MVA-BN Filo received conditional marketing authorization from the EMA in early 2021 as the boosting immunization in a heterologous regimen with adenovirus serotype 26 expressing Ebola virus GP (Zabdeno) [43]. This regimen was well tolerated in clinical trials and generated humoral and cellular responses that persist for up to one year after the boost [100].

Since vaccination for smallpox largely stopped following worldwide eradication in 1980, pre-existing immunity that could impact MVA vector immunogenicity is currently limited to adults over ~40 years old [101]. In animal studies to determine the effect of pre-existing immunity on MVA vector-generated immunity, some reduction in immunogenicity was observed, but this did not impact overall vaccine efficacy [102]. The MVA vector is in use for several clinical stage vaccine candidates for viruses of interest (Table 3).

### 3.3. Paramyxovirus Vectors

Several paramyxoviruses are used as vaccine vectors, including measles virus (MeV), parainfluenza viruses, and Newcastle disease virus (NDV). These vectors can accommodate transgene inserts of 4.5–6 kB, and easily incorporate the membrane GPs of other RNA viruses into their envelope [103,104]. The paramyxovirus vectors discussed here all infect via the intranasal route and are capable of inducing both systemic and local mucosal immune memory [104].

MeV was pursued as a vector based on the success of the live attenuated vaccine for measles, which has a long history as a safe and effective vaccine that generates durable immune memory responses [103]. A large proportion of the human population is seropositive for MeV due to its inclusion in routine childhood immunizations; however, based on animal and human studies, this does not appear to affect the immunogenicity of the MeV vector [105].

The MeV vector vaccines discussed here are based on the MeV Schwarz strain, which was attenuated through serial passage, but retained its ability to replicate and produce viral progeny [106]. A single shot of an MeV vector Lassa virus vaccine (MV-LASV) generated humoral and cellular responses in an NHP study, and was protective when animals were infected, either one month or one year after vaccination [107]. MV-LASV and a MeV vector based Zika virus candidate have completed Phase 1 clinical trials (Table 3).

Parainfluenza virus (PIV) vectors in clinical development include those that utilize human parainfluenza 3 (HPIV3) and parainfluenza virus 5 (PIV5), which, like all PIVs, require a genome length in a multiple of six nucleotides for efficient replication [104].

HPIV3 is a replicating vector that is used as the basis of an Ebola virus vaccine (HPIV3-EbovZ GP) that has completed Phase 1 clinical trials [48]. A unique element of HPIV3-EbovZ GP is that the vaccine was administered intranasally in an NHP study and elicited systemic and local respiratory T cell responses that protected animals from lethal Ebola virus infection [108]. This may be beneficial, as Ebola virus is transmitted via direct contact with infectious body fluids, which may include inoculation via oral or respiratory mucosa [19]. As HPIV3 is a common pediatric respiratory pathogen, the issue of pre-existing immunity is avoided by replacing the native surface proteins of HPIV3 with the Ebola GP [110].

PIV5 is a replicating vector that is currently being assessed in a Phase 1 SARS-CoV-2 trial as an intranasally administered vaccine [49]. PIV5 is a canine pathogen that does not cause disease in humans, but seropositivity is possible if an individual is in regular contact with dogs [111]. However, pre-existing immunity is not a significant concern due to swapping the native PIV surface proteins with that of the target virus; furthermore, a PIV5 vector influenza vaccine proved immunogenic in PIV5 vaccinated canines [111].

NDV is an avian virus that is naturally attenuated for humans, as a key immunomodulatory protein, V protein, is species-specific [112]. NDV is being assessed as a vector for a SARS-CoV-2 vaccine (NDV-HXP-S) that is currently in Phase 1/2 clinical trials using an inactivated form of NDV-HXP-S administered intramuscularly, with or without the Toll-like Receptor 9 agonist adjuvant, CpG 1018 [79]. An interim report of one of the trials found that the vaccine has an acceptable safety profile and is a potent immunogen [113]. A benefit of this vaccine is that it uses egg-based manufacturing, which could make it suitable for large-scale production if existing influenza vaccine manufacturing capacity is leveraged [114].

### 3.4. Vesicular Stomatitis Virus (VSV)

VSV is an attenuated, replicating rhabdovirus vaccine vector that can accommodate antigen inserts of up to ~6 kB in size [28]. For the viruses of interest, the vaccine constructs replace the native VSV surface GP with that of the target virus (Table 3). This serves the purpose of providing the antigenic target, but also attenuates the virus, as GP is a major virulence factor of VSV [115]. The VSV vector is also sometimes further attenuated through the down-regulation of VSV nucleoprotein and the truncation of the native GP cytoplasmic tail [116]. Replacement of the native VSV GP also reduces any potential effects of pre-existing immunity, which is already limited as VSV infection in humans is extremely rare [115].

A VSV vector Ebola virus vaccine (rVSVΔG-ZEBOV-GP) was licensed as ERVEBO by the FDA in 2019 as a single dose vaccine [43]. The effectiveness of this vaccine was established in Phase 3 clinical trials, where robust vaccine efficacy was observed within ten days following the administration of a single dose [26,117]. Vaccine generated anti-GP antibodies are correlated with protection and can persist for at least 24 months [118,119]. A smaller clinical assessment showed that the vaccine also elicits a cellular response [120]. rVSVΔG-ZEBOV-GP has a favorable safety profile and, while some instances of arthralgia and arthritis were reported, they were rare and self-limiting [121]. A SARS-CoV VSV vaccine is currently in a Phase 2b/3 trial (Table 3).

Another rhabdovirus, rabies virus (RABV), is also used as a vaccine vector and similarly can be pseudotyped with the GP of a different target virus, fully replacing the native RABV GP. While there are no RABV vector vaccine candidates in clinical development, the preclinical testing of inactivated vector vaccines for Ebola virus, Sudan virus, Marburg virus, and Lassa virus found the candidates to be immunogenic, either alone or when administered in combination as a multivalent vaccine [122].

## 4. Conclusions and Future Perspectives

Viral vector vaccines have important positive aspects that could be useful in an outbreak, epidemic, or pandemic response. As noted above, such vaccines have played a major role in recent Ebola and COVID-19 responses. These vaccines offer the potential for high levels of efficacy with a single dose, as well as manageable operational logistics. Viral vector vaccines are now in development for several other threats of pandemic potential; however, only a handful of different vectors are being actively pursued for these threats. Understanding the likely safety profile of vectors will inform the selection and development of lead candidates for the next outbreaks or pandemics. Additional investments in a more diverse array of viral vectors are warranted as such vaccines may provide additional tools for future outbreak and pandemic responses. Key factors such as levels of baseline seropositivity against the vector, safety profiles, and development as replication-defective versus live virus vectors should be taken into account when prioritizing likely candidates.

As public and private stakeholders begin to pivot from the COVID-19 response to focus on preparedness for future pandemics, a concerted effort will be required to build a more robust pipeline of vaccines in clinical development, with virus vectors being an important component of that pipeline, in combination with mRNA and protein subunit vaccines. The vaccine community, as a whole, must continue to evaluate lead vectors while investigating the potential for other viral vectors to be progressed into clinical development, and ultimately, to licensed products.

## Figures and Tables

**Table 1 vaccines-10-00436-t001:** Characteristics of select viruses of pandemic potential.

Virus	Fatality Rate	Disease/ Symptoms	Largest Outbreak	Licensed Vaccine
SARS-CoV-2	0.27% [5]	Respiratory illness - acute respiratory distress syndrome (ARDS)	COVID-19 pandemic (as of 31 January 2022) [1] Est. Cases: 373,229,380 Est. Deaths: 5,658,702	Licensed: Yes
SARS-CoV	10% [6]	Respiratory illness - ARDS	2002–2004 SARS pandemic [6] Est. Cases: 8096 Est. Deaths: 774	Licensed: No (Clinical stage candidates [7])
MERS-CoV	34.3% [8]	Respiratory illness - ARDS	2012–2019 sporadic outbreaks [8] Est. Cases: 2499 Est. Deaths: 858	Licensed: No
Ebola virus	50% (25–90% range) [9]	Hemorrhagic fever	2013–2016 West Africa epidemic [10] Est. Cases: 28,616 Est. Deaths: 11,310	Licensed: Yes
Sudan virus	55% [11]	Hemorrhagic fever	2000–2001 Uganda outbreak [11] Est. Cases: 425 Est. Deaths: 224	Licensed: No
Marburg virus	53.8% [12]	Hemorrhagic fever	2004–2005 Angola outbreak [12] Est. Cases: 252 Est. Deaths: 227	Licensed: No
Lassa virus	1–2% [13]	Hemorrhagic fever	Endemic in West Africa 2020 Nigeria cumulative: [14] Est. cases: 6732 Est deaths: 244	Licensed: No
Nipah virus	61.0% [15]	Respiratory symptoms Encephalitis	1998–1999 Malaysia outbreak [16] Est. Cases: 265 Est. Deaths: 105	Licensed: No (Clinical stage candidate [17])
Hendra virus	57% [18]	Respiratory symptoms Encephalitis	Totality of human cases (1994–2008) [17] Est. Cases: 7 Est. Deaths: 4	Licensed: No (Clinical stage candidate for NiV cross-protective for HeV in NHP [16])
Crimean-Congo hemorrhagic fever virus	26.5% [19]	Hemorrhagic fever	Turkey, 2002–2009 period [19] Est. Cases: 4431 Est. Deaths: 222	Licensed: No (Clinical stage candidate)
Rift Valley fever virus	0.5 to 2% [20]	Respiratory symptoms Hemorrhagic fever Encephalitis	2019 Sudan outbreak [21] Est. Cases: 1129 Est. Deaths: 123	Licensed: No
Zika virus	Non-fatal/ 10% for CZS [22]	Fever, arthralgia, maculopapular rash Congenital Zika syndrome (CZS	2015–2016 Zika epidemic (Region of the Americas) Est. Cases: 707,133 [23] Est. Deaths due to CZS (Brazil): 603/6059 CZS cases [22]	Licensed: No

**Table 2 vaccines-10-00436-t002:** Target product profile considerations.

Criteria	Objective
Indication for use	For active immunization of persons considered at-risk of exposure; reactive use in response to outbreaks may be preferrable
Target population	All adults and pediatrics down to 6 months of age
Safety/Reactogenicity	Safety and reactogenicity that provide a favorable benefit-risk profile in context with vaccine efficacy; ideally only mild, transient vaccination-related adverse events (AE) and no vaccine-related serious AEs (SAEs)
Efficacy (clinical)	Greater than 90% efficacy in preventing infection or disease in healthy adults (70% minimum). If demonstration of clinical efficacy is not feasible, pre-clinical immunogenicity and efficacy in a standardized and relevant animal model together with clinical immunogenicity may be considered.
Efficacy (nonclinical based on Animal Rule)	Demonstration of protection in relevant animal models in line with FDA Animal Rule guidance
Onset to protection	Rapid onset to protection within two weeks after first dose
Duration of protection	Primary series confers long-lasting protection of 1 year or more, and can be maintained by booster doses
Dosing regimen	Single-dose primary series
Route of administration	Injectable (IM, ID, or SC) using standard volumes suitable for a single injection, but oral or other needle-free approaches would be preferred.
Storage temperature	Room temperature > 2–8 °C > −20 °C

**Table 3 vaccines-10-00436-t003:** Approved and clinical-stage viral vector vaccine candidates for viruses of interest.

Replication	Vector	Target	Vaccine	Status
Replicating	VSV	SARS-CoV-2	VSV-ΔG SARS-CoV-2	Phase 2/3 active [41]
		MERS-CoV-2	BVRS-GamVac-Combi (Ad5 & VSV)	Phase 1/2 active [42]
		EBOV	rVSVΔG-ZEBOV-GP	Approved [43]
			GamEvac-Combi & GamEvac-Lyo (Ad5 & VSV)	Licensed (Russia) [43]
		LASV	rVSV∆G-LASV-GPC	Phase 1 active [44]
	MeV	LASV	MV-LASV	Phase 1 complete [45]
		ZIKV	MV-ZIKA & MV-ZIKA-RSP	Phase 1 complete [46,47]
	HPIV3	EBOV	HPIV3-EbovZ GP	Phase 1 complete [48]
	PIV5	SARS-CoV-2	CVXGA1-001	Phase 1 active [49]
	LAIV	SARS-CoV-2	DelNS1-2019-nCoV-RBD-OPT1	Phase 3 active [50]
Non-replicating	Ad5	SARS-CoV-2	Ad5-nCoV	Authorized (China) [51]
			VXA-Cov2-1	Phase 2 active [52]
			hAd5-S-Fusion+N-ETSD	Phase 1/2 active [53]
			Ad5-triCoV/Mac	Phase 1 active [54]
		MERS-CoV	MERS BVRS-GamVac	Phase 1/2 active [55]
			MERS BVRS-GamVac-Combi (Ad5 & VSV)	Phase 1/2 active [42]
		EBOV	Ad5-EBOV	Licensed (China) [43]
			GamEvac-Combi & Lyo (Ad5 & VSV)	Licensed (Russia) [43]
	Ad26	SARS-CoV-2	Ad26.COV2.S	Authorized [56,57]
			Sputnik Light	Approved (Russia) [58]
		EBOV	Ad26.ZEBOV-GP (in Zabdeno/Mvabea vaccine)	Authorized [43]
		ZIKV	Ad26.ZIKV.001	Phase 1 complete [59]
	ChAdOx1	SARS-CoV-2	ChAdOx1-S	Authorized [60]
		MERS-CoV	ChAdOx1 MERS	Phase 1 complete [61]
		EBOV/SUDV	ChAdOx1 biEBOV	Phase 1 active [62]
		RVFV	ChAdOx1 RVF	Phase 1 active [63]
		ZIKV	ChAdOx1 Zika	Phase 1 active [64]
	CAd3	EBOV	ChAd3-EBO-Z	Phase 2 complete [65]
		MARV	cAd3-MARV	Phase 1 complete [66,67]
		SUDV	cAd3-EBO S	Phase 1 complete [67]
		EBOV/SUDV	cAd3-EBO	Phase 1 complete [68]
	ChAd68	SARS-CoV-2	ChAdV68-S	Phase 1 active [69]
	Ad6 single cycle	SARS-CoV-2	SC-Ad6-1	Phase 1 active [70]
	Ad36	SARS-CoV-2	BBV154	Phase 1 active [71]
	Ad5/35	SARS-CoV-2	AdCLD-CoV19	Phase 1/2 active [72]
	GRAd32	SARS-CoV-2	GRAd-COV2	Phase 2/3 active [73]
	AAV5	SARS-CoV-2	AAV5-RBD-S	Phase 1/2 active [74]
	MVA	SARS-CoV-2	MVA-SARS-2-S	Phase 1 complete [75]
			MVA-SARS2-ST	Phase 1 active [76]
			COH04S1	Phase 2 active [77]
		MERS-CoV-2	MVA-MERS-S	Phase 1 complete [78]
		Filovirus multivalent	MVA-BN-Filo (in Zabdeno/Mvabea vaccine)	Authorized [43]
	NDV	SARS-CoV-2	NDV-HXP-S +/− CpG 1018	Phase 1/2 active [79]

## Data Availability

Not applicable.

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
