# Peer review of "Utilization of Viral Vector Vaccines in Preparing for Future Pandemics"

_vaccines, 2022, doi:10.3390/vaccines10030436_

Round 1
Reviewer 1 Report
In this study, the authors are discussing some of the advantages offered by vaccines utilizing virus vectors some in the context of preparing for future pandemics: viral-vectored vaccines may provide opportunities and. could be useful in an outbreak, epidemic, or pandemic response, to slow the spread of disease.
The report is well supported by other
published material based on a favorable balance between the benefits of vaccine effectiveness, the seriousness of the threat and a comparably favorable safety profile.
The conclusions are enough consistent with the evidence and
arguments presented.
The references result appropriate.
The tables must be well elaborated.
Minor comments:
L 32: reports the full name of the viruses SARS-CoV-2, MERS-CoV, SARS-CoV-1
L 35: explain better the meaning of the sentence “… have been published by GAVI and CEPI”
L56-57: explain better the meaning of the sentence “with antibodies only peaking following recovery from disease”
L61: Better “a key”
L64-65 explain better the sentence: “to intervene with vaccines prior to the event becoming an epidemic or pandemic”.
L94: reports the full name of US.
L113:-70°C instead of -70C.
L140: add a reference after “by their robust induction of innate and adaptive immune responses”.
L187: reports the full name of “ChAdOx1”.
L197: reports the full name of “AZD1222”.
L242: reports the full name of “Ad26-ZEBOV”.
L247: delete “(Taub 2008)”.
L256: report the full name of “GPs”.
L318: better “is also used as a vaccine….”
Table 1: reduce the format of the table 1 and set a single character.
Table 3: Set a single font size for all the characters.
Author Response
Thank you for the comments. We believe we were able to address all of them.
Dan
Thank you for the quick review and constructive comments on our draft manuscript. We’ve made revisions to the manuscript as noted below in bold text, but please let us know if you have any further questions.
Kimberly Hofmeyer, Katherine Bianchi, and Daniel Wolfe
L 32: reports the full name of the viruses SARS-CoV-2, MERS-CoV, SARS-CoV-1. Agree, we have corrected with the full name.
L 35: explain better the meaning of the sentence “… have been published by GAVI and CEPI” We noted “to their respective websites” and references 3 and 4 are for their websites.
L56-57: explain better the meaning of the sentence “with antibodies only peaking following recovery from disease” The statement was changed to “For Lassa virus, human and animal studies indicate that convalescence is largely me-diated by the cellular response. Antibody titers with antibodies only are highest only after infected individuals have already recovered peaking following recovery from disease.”
L61: Better “a key” Agree, that change has been made
L64-65 explain better the sentence: “to intervene with vaccines prior to the event becoming an epidemic or pandemic”. Noted that with diseases that are geographically isolated at the early stages of an outbreak, we can intervene prior to a pandemic stage
L94: reports the full name of US. Agree, specified United States
L113:-70°C instead of -70C. Agree, thanks for catching
L140: add a reference after “by their robust induction of innate and adaptive immune responses”. Agree, reference added.
L187: reports the full name of “ChAdOx1”. ChAdOx1 is more of a designation from the company/academic partnership between Oxford and AstraZeneca. It stands for Chimpanzee Adenovirus Oxford, but it may be best to not spell that out and imply “Oxford” is part of the name of a naturally occurring virus.
L197: reports the full name of “AZD1222”. This is the code name applied to AstraZeneca’s vaccine.
L242: reports the full name of “Ad26-ZEBOV”. Agree, the full name is included.
L247: delete “(Taub 2008)”. Agree, thanks for catching.
L256: report the full name of “GPs”. Agree, it should be glycoproteins.
L318: better “is also used as a vaccine….” Agreed, thank you.
Table 1: reduce the format of the table 1 and set a single character. The Table 3 format has been reduced and uses a single character type.
Table 3: Set a single font size for all the characters. Table 3 has been set to have a single font size.
Reviewer 2 Report
This is an interesting review dealing with the use of viral-vectored vaccines as a weapon against viral threat with pandemic potential. No doubt, the area is very topical, important and relevant. It is generally well written, and my comments aimed to increase the scientific soundness and clarity of it.
- The topic of this article is not new and has been presented several times before. There are other comprehensive articles dealing with the same subject (see below). To make this article more interesting to a reader I suggest the authors to precisely and clearly define the target question of this review.
See:
a) Ghattas M, Dwivedi G, Lavertu M, Alameh MG. Vaccine Technologies and Platforms for Infectious Diseases: Current Progress, Challenges, and Opportunities. Vaccines (Basel). 2021 Dec 16;9(12):1490. doi: 10.3390/vaccines9121490. PMID: 34960236; PMCID: PMC8708925.
b) Vrba SM, Kirk NM, Brisse ME, Liang Y, Ly H. Development and Applications of Viral Vectored Vaccines to Combat Zoonotic and Emerging Public Health Threats. Vaccines (Basel). 2020 Nov 13;8(4):680. doi: 10.3390/vaccines8040680. PMID: 33202961; PMCID: PMC7712223.
c) Pollard AJ, Bijker EM. A guide to vaccinology: from basic principles to new developments. Nat Rev Immunol. 2021 Feb;21(2):83-100. doi: 10.1038/s41577-020-00479-7. Epub 2020 Dec 22. Erratum in: Nat Rev Immunol. 2021 Jan 5;: PMID: 33353987; PMCID: PMC7754704.
- Chapter 4 should be “Conclusions and Future Perspectives” but not “Summary”. Summary was already presented in line 9
- “Simple summary” is missing.
- The number of references is impressive. However, in my opinion it can be limited, especially since many of them refer to the same aspects.
Author Response
Thank you for the quick review and constructive comments on our draft manuscript. We’ve made revisions to the manuscript as noted below, but please let us know if you have any further questions.
Kimberly Hofmeyer, Katherine Bianchi, and Daniel Wolfe
This is an interesting review dealing with the use of viral-vectored vaccines as a weapon against viral threat with pandemic potential. No doubt, the area is very topical, important and relevant. It is generally well written, and my comments aimed to increase the scientific soundness and clarity of it.
- The topic of this article is not new and has been presented several times before. There are other comprehensive articles dealing with the same subject (see below). To make this article more interesting to a reader I suggest the authors to precisely and clearly define the target question of this review. We agree, especially with the manuscript from Vrba, et.al. Clearly defining the target question is an excellent suggestion and we’ve proposed adding the following sentence at the end of the introduction section “A recent review by Vrba, et.al. highlighted the breadth of viruses that have been used to express heterologous antigens as a means for delivery as a vaccine (will insert reference from Vrba). In this review, we will focus on those virus vectors that have proceeded to clinical development, and how these vectors could be applied to emerging viral threats of pandemic potential.”
See:
- a) Ghattas M, Dwivedi G, Lavertu M, Alameh MG. Vaccine Technologies and Platforms for Infectious Diseases: Current Progress, Challenges, and Opportunities. Vaccines (Basel). 2021 Dec 16;9(12):1490. doi: 10.3390/vaccines9121490. PMID: 34960236; PMCID: PMC8708925. This was a very good review of vaccine advances across all platforms, and goes into greater discussion of mechanisms of action for vaccines in general and each major class of vaccines (recombinant protein, DNA, viral vectors, etc.) where this review is focused on viral vectors and dives deeper into the various vectors in development.
- b) Vrba SM, Kirk NM, Brisse ME, Liang Y, Ly H. Development and Applications of Viral Vectored Vaccines to Combat Zoonotic and Emerging Public Health Threats. Vaccines (Basel). 2020 Nov 13;8(4):680. doi: 10.3390/vaccines8040680. PMID: 33202961; PMCID: PMC7712223. This review has the most overlap with our current draft and will be referenced. The biggest difference we see is that our current draft may be a logical build on this review, where we focus on those vectors in clinical development and how they may be applied to diseases of pandemic potential.
- c) Pollard AJ, Bijker EM. A guide to vaccinology: from basic principles to new developments. Nat Rev Immunol. 2021 Feb;21(2):83-100. doi: 10.1038/s41577-020-00479-7. Epub 2020 Dec 22. Erratum in: Nat Rev Immunol. 2021 Jan 5;: PMID: 33353987; PMCID: PMC7754704. This is a much more comprehensive overview of vaccinology and vaccine history as a whole.
- Chapter 4 should be “Conclusions and Future Perspectives” but not “Summary”. Summary was already presented in line 9 Agree, we have made that change.
- “Simple summary” is missing. I’m not sure we understand the request. We might have a question for the editors in this case, is this something that’s usually provided for research articles? If needed for a review, we can provide one. We could include the following as a simple summary. “The use of virus vectors in vaccine development has progressed rapidly over the past decade with applications to Ebola virus and SARS-CoV-2 among other diseases. This class of vaccines has potential benefits in terms of the immune response that is elicited and use in outbreak settings. In this review, we will discuss the virus vectors that have progressed into clinical development, and some of the pros and cons of applying these approaches to emerging infectious diseases.”
- The number of references is impressive. However, in my opinion it can be limited, especially since many of them refer to the same aspects. The following references were removed and consolidated under other remaining references:
Huttner, A.; Dayer, J.A.; Yerly, S.; Combescure, C.; Auderset, F.; Desmeules, J.; Eickmann, M.; Finckh, A.; Goncalves, A.R.; Hooper, J.W.; et al. The effect of dose on the safety and immunogenicity of the VSV Ebola candidate vaccine: a randomised double-blind, placebo-controlled phase 1/2 trial. Lancet Infect Dis 2015, 15, 1156-1166, doi:10.1016/s1473-3099(15)00154-1.
Humphreys, I.R.; Sebastian, S. Novel viral vectors in infectious diseases. Immunology 2018, 153, 1-9, doi:10.1111/imm.12829.
Logunov, D.Y.; Dolzhikova, I.V.; Shcheblyakov, D.V.; Tukhvatulin, A.I.; Zubkova, O.V.; Dzharullaeva, A.S.; Kovyrshina, A.V.; Lubenets, N.L.; Grousova, D.M.; Erokhova, A.S.; et al. Safety and efficacy of an rAd26 and rAd5 vector-based heterologous prime-boost COVID-19 vaccine: an interim analysis of a randomised controlled phase 3 trial in Russia. Lancet 2021, 397, 671-681, doi:10.1016/s0140-6736(21)00234-8.
Barouch, D.H.; Kik, S.V.; Weverling, G.J.; Dilan, R.; King, S.L.; Maxfield, L.F.; Clark, S.; Ng'ang'a, D.; Brandariz, K.L.; Abbink, P.; et al. International seroepidemiology of adenovirus serotypes 5, 26, 35, and 48 in pediatric and adult populations. Vaccine 2011, 29, 5203-5209, doi:10.1016/j.vaccine.2011.05.025.
European Medicines Agency. Zabdeno. Available online: https://www.ema.europa.eu/en/medicines/human/EPAR/zabdeno (accessed on 14 December 2021).
Afolabi, M.O.; Ishola, D.; Manno, D.; Keshinro, B.; Bockstal, V.; Rogers, B.; Owusu-Kyei, K.; Serry-Bangura, A.; Swaray, I.; Lowe, B.; et al. Safety and immunogenicity of the two-dose heterologous Ad26.ZEBOV and MVA-BN-Filo Ebola vaccine regimen in children in Sierra Leone: a randomised, double-blind, controlled trial. Lancet Infect Dis 2022, 22, 110-122, doi:10.1016/s1473-3099(21)00128-6.
Folegatti, P.M.; Bittaye, M.; Flaxman, A.; Lopez, F.R.; Bellamy, D.; Kupke, A.; Mair, C.; Makinson, R.; Sheridan, J.; Rohde, C.; et al. Safety and immunogenicity of a candidate Middle East respiratory syndrome coronavirus viral-vectored vaccine: a dose-escalation, open-label, non-randomised, uncontrolled, phase 1 trial. Lancet Infect Dis 2020, 20, 816-826, doi:10.1016/S1473-3099(20)30160-2.
Ledgerwood, J.E.; DeZure, A.D.; Stanley, D.A.; Coates, E.E.; Novik, L.; Enama, M.E.; Berkowitz, N.M.; Hu, Z.; Joshi, G.; Ploquin, A.; et al. Chimpanzee Adenovirus Vector Ebola Vaccine. N Engl J Med 2017, 376, 928-938, doi:10.1056/NEJMoa1410863.
Crosby, C.M.; Nehete, P.; Sastry, K.J.; Barry, M.A. Amplified and persistent immune responses generated by single-cycle replicating adenovirus vaccines. J Virol 2015, 89, 669-675, doi:10.1128/jvi.02184-14.
Pittman, P.R.; Hahn, M.; Lee, H.S.; Koca, C.; Samy, N.; Schmidt, D.; Hornung, J.; Weidenthaler, H.; Heery, C.R.; Meyer, T.P.H.; et al. Phase 3 Efficacy Trial of Modified Vaccinia Ankara as a Vaccine against Smallpox. N Engl J Med 2019, 381, 1897-1908, doi:10.1056/NEJMoa1817307.
Volz, A.; Sutter, G. Modified Vaccinia Virus Ankara: History, Value in Basic Research, and Current Perspectives for Vaccine Development. Adv Virus Res 2017, 97, 187-243, doi:10.1016/bs.aivir.2016.07.001.
Kannanganat, S.; Nigam, P.; Velu, V.; Earl, P.L.; Lai, L.; Chennareddi, L.; Lawson, B.; Wilson, R.L.; Montefiori, D.C.; Kozlowski, P.A.; et al. Preexisting vaccinia virus immunity decreases SIV-specific cellular immunity but does not diminish humoral immunity and efficacy of a DNA/MVA vaccine. J Immunol 2010, 185, 7262-7273, doi:10.4049/jimmunol.1000751.
Knuchel, M.C.; Marty, R.R.; Morin, T.N.; Ilter, O.; Zuniga, A.; Naim, H.Y. Relevance of a pre-existing measles immunity prior immunization with a recombinant measles virus vector. Hum Vaccin Immunother 2013, 9, 599-606, doi:10.4161/hv.23241.
U.S. Food & Drug Administration. ERVEBO. Available online: https://www.fda.gov/vaccines-blood-biologics/ervebo
European Medicines Agency. Ervebo. Available online: https://www.ema.europa.eu/en/medicines/human/EPAR/ervebo (accessed on 14 December 2021).
University of Oxford. Safety and Immunogenicity of a Candidate RVFV Vaccine (RVF001). Available online: https://ClinicalTrials.gov/show/NCT04754776 (accessed on 14 December 2021).
University of Oxford. Safety and Immunogenicity of a Candidate ZIKV Vaccine (ZIKA001). Available online: https://ClinicalTrials.gov/show/NCT04015648 (accessed on 14 December 2021).